# Protein Profiling of Breast Carcinomas Reveals Expression of Immune-Suppressive Factors and Signatures Relevant for Patient Outcome

**DOI:** 10.3390/cancers14184542

**Published:** 2022-09-19

**Authors:** Felix Ruoff, Nicolas Kersten, Nicole Anderle, Sandra Jerbi, Aaron Stahl, André Koch, Annette Staebler, Andreas Hartkopf, Sara Y. Brucker, Markus Hahn, Katja Schenke-Layland, Christian Schmees, Markus F. Templin

**Affiliations:** 1NMI Natural and Medical Sciences Institute at the University of Tuebingen, 72770 Reutlingen, Germany; 2FZI Research Center for Information Technology, Intelligent Systems and Production Engineering (ISPE), 76131 Karlsruhe, Germany; 3Interfaculty Institute for Biomedical Informatics (IBMI), University of Tuebingen, 72076 Tuebingen, Germany; 4Department of Women’s Health, University of Tuebingen, 72076 Tuebingen, Germany; 5Institute of Pathology and Neuropathology, University of Tuebingen, 72076 Tuebingen, Germany; 6Department of Women’s Health, University of Ulm, 89081 Ulm, Germany; 7Cluster of Excellence iFIT (EXC2180) “Image-Guided and Functionally Instructed Tumor Therapies”, University of Tuebingen, 72076 Tuebingen, Germany; 8Institute of Biomedical Engineering, Department for Medical Technologies and Regenerative Medicine, University of Tuebingen, 72076 Tuebingen, Germany

**Keywords:** DigiWest, breast cancer, cellular signaling, PPARγ, immune cell infiltration, patient stratification

## Abstract

**Simple Summary:**

Breast cancer treatment has improved substantially over the last decade. Still, the failure of treatment and therapy resistance are urgent problems. Here, we assessed cellular signaling within primary cancer tissue to evaluate the possibility of developing strategies that lead to better patient stratification and the development of personalized treatment options. By employing DigiWest technology, the expression and activation of the regulators of key signaling pathways in breast cancer tissue were monitored. A positive correlation between immune cell infiltration and event-free survival was detected. PPARγ activation showed a negative correlation with immune cell infiltration, suggesting a novel immune evasion mechanism.

**Abstract:**

In cancer, the complex interplay between tumor cells and the tumor microenvironment results in the modulation of signaling processes. By assessing the expression of a multitude of proteins and protein variants in cancer tissue, wide-ranging information on signaling pathway activation and the status of the immunological landscape is obtainable and may provide viable information on the treatment response. Archived breast cancer tissues from a cohort of 84 patients (no adjuvant therapy) were analyzed by high-throughput Western blotting, and the expression of 150 proteins covering central cancer pathways and immune cell markers was examined. By assessing CD8α, CD11c, CD16 and CD68 expression, immune cell infiltration was determined and revealed a strong correlation between event-free patient survival and the infiltration of immune cells. The presence of tumor-infiltrating lymphocytes was linked to the pronounced activation of the Jak/Stat signaling pathway and apoptotic processes. The elevated phosphorylation of PPARγ (pS112) in non-immune-infiltrated tumors suggests a novel immune evasion mechanism in breast cancer characterized by increased PPARγ phosphorylation. Multiplexed immune cell marker assessment and the protein profiling of tumor tissue provide functional signaling data facilitating breast cancer patient stratification.

## 1. Introduction

Breast cancer is the fifth most common cause of cancer-related deaths worldwide, with 2.2 million new cases and around 685,000 deaths in 2020 [1]. A variety of therapeutic options, including surgery, chemo-, hormone, and biological therapies, are available, and survival rates have increased substantially over the years [2]. Nevertheless, current cancer therapies lack a more personalized approach, and long-term therapy resistance has become a focus of current research [3,4]. Novel insights into the establishment, interaction and control of the tumor immune microenvironment (TIME) have drawn a more comprehensive picture of factors that might account for treatment failure or severe side effects [3,5]. The detection of tumor-infiltrating lymphocytes (TIL) is seen as a major prognostic factor in different carcinomas and has been suggested as a routine pathological evaluation [6,7]. In general, infiltrating immune cells can be categorized as either (i) pro-tumoral or (ii) anti-tumoral [6]. Some leukocytes, such as type 2 macrophages and regulatory T-cells (Tregs), are associated with pro-tumoral effects [8,9], whereas type 1 macrophages, effector T-cells, natural killer (NK) cells and dendritic cells (DC) are linked to anti-tumoral effects [10,11]. However, the complex interplay among different simultaneous immune responses and the way that cancer cells can modulate them are not yet fully understood. Large investigations on the bulk gene expression of infiltrating immune cells have been performed that associated different immune cell types with the risk of relapse [12,13]. Higher infiltration of immune cells has been linked to patient outcomes and treatment responses in several studies [8,14,15,16]. For example, the presence of cytotoxic T-cells (CD8+) has been associated with better survival in estrogen receptor (ER)-negative and ER-positive/human epidermal growth factor receptor 2 (Her2)-positive breast cancer subtypes [14]. Additionally, the categorization of tumors into immunological inflamed (hot) and non-inflamed (cold) cancers has been advocated for certain tumor entities [17,18]. Most studies focus on the evaluation of single immune cell markers or solely investigate the degree of TIL infiltration, most commonly using imaging methods (H&E/IHC/IF staining) for immune cell assessment and subsequent scoring [19,20]. Yet, recent studies have shown that classification based on one marker may be an oversimplification and does not entirely represent the complex nature of the immune response against cancer [21,22]. In the present study, we established the multiplexed assessment of several immune cell markers by DigiWest [23], a multiplexed bead-based Western blot, in fresh-frozen tissue samples.

The semi-quantitative protein expression analysis of primary breast cancer tissue allowed for the detection of infiltrating immune cells and the concomitant monitoring of the activation state of central signaling pathways. The activation of immune cell signaling and the induction of apoptotic processes were observed in highly immune-cell-infiltrated tumor tissue, whereas the activation of PPAR gamma signaling was found in tumors with low-level immune cell infiltration.

## 2. Materials and Methods

### 2.1. Patient Cohort

A retrospective cohort of primary unilateral invasive mamma carcinomas from patients who underwent a primary resection was utilized (snap-frozen, *n* = 159, tumor bank University Hospital Tuebingen). Inclusion criteria were hormone-receptor- and/or human epidermal growth factor receptor (Her2)-negative or -positive carcinomas, determined by immunohistochemistry at the Institute of Pathology, University of Tuebingen, Germany, at the time of surgery. Samples were further classified by the occurrence of distant metastases or local relapse within 10 years (poor responder) versus no occurrence of distant metastases, or a local relapse within 10 years or contralateral carcinoma within 5 years of the primary diagnosis (good responder). In general, the exclusion criteria were the occurrence of contralateral mamma carcinomas before the occurrence of distant metastases in the poor-responder subgroup and within 5 years in the good-responder subgroup, as well as the presence of bilateral mamma carcinomas or other malignancies (Appendix A). All patients enrolled neither received any neo-adjuvant treatment nor had any known metastases before surgery.

### 2.2. Sample Preparation and Assessment of Tumor Content

Layered cuts of each fresh-frozen sample were prepared, and Hematoxylin–Eosin (H&E) staining was performed according to standard protocols for the first and second layers. Between layers, 100 µm of tissue was trimmed, collected and stored at −80 °C. Prior to protein expression analysis, the prepared layered cuts were H&E-stained (see also Figure 1A), and sections 100 µm apart were re-assessed by a pathologist (A.S). The evaluation of these sections revealed that 2.5% (*n* = 4) of the samples were normal tissue, 5.0% (*n* = 8) were ductal carcinoma in situ (DCIS), and 1.9% (*n* = 3) mostly contained necrotic tissue. In addition, 90.6% (*n* = 144) were classified as invasive ductal carcinoma (IDC) (see also Figure 1B). Yet, 2.5% (*n* = 4) showed a tumor content of app. 5–10%, and 27.7% (*n* = 44) showed a tumor content in between 15% and 45%. Of all samples, 60.4% (*n* = 96) showed a tumor content of 50% or higher (up to 95%) (see also Figure 1B). Samples with ≥50% tumor content were selected for further analysis (*n* = 84), and intermediate sections from the generated layered cuts were used for protein preparation. In addition, *n* = 10 samples classified as normal tissue or with low tumor content (>10%) were assigned to the baseline control group. Samples were lysed by incubating the collected tissue at 95 °C for 10 min in lysis buffer (4% LDS, 50 mM DTT) (Appendix A).

To confirm the tumor content of the enrolled sample set, the abundances of the proliferation marker Ki-67 and general carcinoma markers Cytokeratin 8/18, Cytokeratin 8 (pS23) and Cytokeratin 6 were assessed by DigiWest analysis (see below). The expression of these markers was found to be significantly different between the tumor sample set and baseline sample sets, revealing a high tumor content in the analysis of the former sample set (Appendix A).

### 2.3. Compliance of Receptor Status

To review the compliance of the prepared samples with the pathological evaluation, the expression levels of the hormone receptors estrogen receptor (ER) and progesterone receptor (PR) and human Her2 were analyzed, and the resulting signals were compared to the pathological receptor status. A significant difference (*p* < 0.05, Mann–Whitney U test) in the signal was found between samples pathologically classified as receptor-positive and receptor-negative or the baseline group. Samples were categorized into three groups by referencing the pathological receptor status. In ER/PR− Her2− samples (*n* = 18), low or no expression of ER/PR or Her2 was observed. The analysis of ER/PR+ Her2− samples (*n* = 45) showed the increased expression of ER and a slight increase in expression in PR but not in Her2, whereas ER/PR+/− Her2+ samples (*n* = 20) displayed a significant increase in Her2 expression compared to the other groups (see also Figure 1C,D; *p* < 0.05, Mann–Whitney U test). We concluded that the DigiWest measurement of hormone receptors and Her2 expression is comparable to the classical pathological assessment of receptor status in the present cohort.

### 2.4. Immunohistochemical Staining

Immunohistochemical staining was performed on 5 µm formalin-fixed paraffin-embedded sections. After de-paraffinization, epitope retrieval was performed at 95 °C for 20 min in the appropriate antigen retrieval buffer. BLOXXALL-Blocking solution (Vector Laboratories, Burlingame, CA, USA) was added for 10 min. After washing in PBS, the sections were incubated with blocking buffer (PBS, 0.25% Triton-X-100, 10% goat serum, 4 drops/mL streptavidin (Vector Laboratories)). Primary antibody diluted in dilution buffer (PBS, 1%BSA, Biotin (Vector Laboratories)) was added and incubated in a humidified chamber. Rabbit (rb) anti-CD8α (#85336, clone D8A8Y, Cell Signaling Technology (CST), Leiden, Netherlands, 1:100 dilution), rb anti-CD11c (#45581, clone D3V1E, CST, 1:400 dilution), rb anti-CD68 (#76437, MultiMab, CST, 1:200 dilution) and rb anti-CD16 (ab24622, clone EPR14336, Abcam, 1:400 dilution) antibodies were used for staining. The appropriate biotin-conjugated secondary antibody (Jackson Immuno Research, Cambridge, UK) diluted in PBS/1%BSA was added for 30 min. After subsequent washing in PBST, slides were incubated with streptavidin-labeled horseradish peroxidase. Peroxidase activity was developed with Novolink 3,3′-Diaminobenzidine (Leicabiosystems, Nußloch, Germany). Slides were counterstained with Hematoxylin QS (Vector Laboratories).

Staining for PR, PPARγ and PPARγ–pS112 was performed using a DAKO Autostainer Link 48 (Dako, Jena, Germany), and antigen retrieval was performed using a DAKO PT Link (Dako) according to the manufacturer’s recommendations. For staining with mouse anti-PR antibody (IR06861, clone PgR636, Dako), slides were incubated in FLEX TRS HIGH pH buffer (K8004, Dako) at 85 °C for 20 min, followed by primary antibody incubation for 20 min and incubation with a mouse linker for 30 min. Subsequently, slides were incubated with a universal secondary antibody (EnVision FLEX/HRP, K8000, Dako) for 15 min. For staining with rb anti-PPARγ (#2435, clone C26H12, CST, dilution 1:100) and rb anti-PPARγ–pS112 (orb5574, Biorbyt, dilution 1:400), slides were incubated in FLEX TRS LOW pH buffer (K8005, Dako) at 85 °C for 20 min, followed by primary antibody incubation for 30 min and universal secondary antibody incubation for 20 min. For detection, the EnVision detection system (K500711-2, Dako) was used.

Whole-slide images were taken utilizing an Axio Scan Z.1 (Zeiss, Oberkochen, Germany). For the evaluation of staining intensity, five representative sections of each slide were used, and the mean intensity of DAB staining in positive pixel^2^ was calculated utilizing ZenBlue software 3.1 (Zeiss).

### 2.5. Multiplex Protein Profiling Via DigiWest

DigiWest was performed as described previously [23]. Briefly, the NuPAGE system (Life Technologies, Carlsbad, CA, USA) with a 4–12% Bis-Tris gel was used for gel electrophoresis and Western blotting onto PVDF membranes. After washing with PBST, proteins were biotinylated by adding 50 µM NHS-PEG12-Biotin in PBST for 1 h to the membrane. After washing in PBST, membranes were dried overnight. Each Western blot lane was cut into 96 strips of 0.5 mm each. Strips of one Western blot lane were sorted into a 96-well plate (Greiner Bio-One, Frickenhausen, Germany) according to their molecular weights. Protein elution was performed using 10 µL of elution buffer (8 M Urea and 1% Triton-X100 in 100 mM Tris-HCl pH 9.5). Neutravidin-coated MagPlex beads (Luminex, Austin, TX, USA) of a distinct color ID were added to the proteins of a distinct molecular weight fraction, and coupling was performed overnight. Leftover binding sites were blocked by adding 500 µM deactivated NHS-PEG12-Biotin for 1 h. To reconstruct the original Western blot lane, the beads were pooled, for which the color IDs represent the molecular weight fraction of the proteins.

For antibody incubation, 5 µL of the DigiWest Bead mixes were added to 50 µL of assay buffer (Blocking Reagent for ELISA (Roche, Rotkreuz, Switzerland) supplemented with 0.2% milk powder, 0.05% Tween-20 and 0.02% sodium azide) in a 96-well plate. Assay buffer was discarded, and 30 µL of primary antibody diluted in assay buffer was added per well. Primary antibodies were incubated overnight at 15 °C on a shaker. Subsequently, they were washed twice with PBST. After washing, 30 µL of species-specific secondary antibody diluted in assay buffer labeled with phycoerythrin was added, and incubation took place for 1 h at 23 °C. Before the readout on a Luminex FlexMAP 3D instrument, the beads were washed twice with PBST.

Analyses and peak integration were performed by utilizing the novel DigiWest-Analyzer software package [24].

### 2.6. Statistical Analysis

Statistical comparison was performed by using the Mann–Whitney U test (GraphPad Prism version 9.2.0, GraphPad Software, San Diego, CA, USA). A Spearman correlation analysis, hierarchical cluster analysis, Chi-square test, Kaplan–Meier plot and log-rank test were carried out utilizing the DigiWest-Evaluator software package [24]. *p* values of <0.05 were considered statistically significant if not stated differently.

### 2.7. Pathway Enrichment Analysis

Testing for significantly enriched pathways was performed with an over-representation analysis using Fisher’s exact test with subsequent calculation of Storey’s Q-values for multiple testing correction. The subsets of analytes that were used for this analysis were defined by applying the Mann–Whitney U test to identify differentially expressed analytes between the good-responder and poor-responder groups. The pathway enrichment pipeline was carried out utilizing the DigiWest-Evaluator software package [24].

## 3. Results

### 3.1. Sample Quality Control and DigiWest Protein Expression Analysis

After the initial sample assessment (*n* = 159, Figure 1A,B), samples with a tumor content >50% and a sufficient protein amount (*n* = 84), as well as control samples with 10% or less tumor content (*n* = 10), were selected for extensive protein expression analysis. To identify markers relevant for the differentiation of good and poor responders, we measured 150 proteins and protein variants using DigiWest, mainly focusing on functional signal transduction, i.e., protein phosphorylation (covering 41 phospho-variants). This extensive expression analysis encompassed cell-cycle-control, apoptosis, Jak/Stat, MAPK, Pi3K/Akt, Wnt and autophagic signaling pathways, as well as general tumor and immune cell markers. The DigiWest evaluation of hormone receptor and Her2 receptor expression complied with the pathologically assessed receptor status, confirming the high quality of the selected tumor samples (Figure 1C,D).

To examine the connection between cellular signaling and the responder status, PANTHER pathway enrichment analysis was conducted for all proteins differentially expressed between good- and poor-responder samples [25,26]. The highest –log2 Q-value was found for the Jak/Stat signaling pathway (Figure 2A). Additionally, members of Jak/Stat signaling and several immune cell markers displayed significant differences when comparing the protein expression of good- and poor-responder samples (Mann–Whitney U test, FDR limit 0.1, Figure 2B). Taken together, these results indicate a connection between immune-cell-related signaling pathway activity and patient treatment response.

### 3.2. Patient Stratification Based on Immune Cell Infiltration Analysis by DigiWest

The degree and type of tumor infiltration by immune cells have become a novel and promising stratification factor for evaluating patient outcomes. Therefore, we evaluated the subset of measured immune cell markers in more detail. A correlation analysis of CD8α, CD4, CD68, CD11c, CD16, CD56, CD25 and CD163 protein expression was performed. CD8α, CD68, CD11c and CD16 displayed the highest correlation (Spearman’s r < 0.55, Figure 3A, Appendix A), suggesting the co-occurrence of represented immune cells.

Unsupervised hierarchical cluster linkage analysis of CD8α (a common marker for cytotoxic T-cells), CD16 (a common marker for cytotoxic natural killer (NK) cells), CD11c (a marker for dendritic cells) and CD68 (a general marker for macrophages) revealed two distinct sample groups with different levels of immune marker expression (Euclidean distance, complete linkage, Figure 3B,C). The group with higher immune cell marker expression (*n* = 27) is referred to as “hot tumors”, whereas the group with lower immune cell marker expression (*n* = 57) is referred to as “cold tumors”.

Subsequently, CD8α, CD68, CD11c and CD16 were immunohistochemically stained on matched FFPE sections, when available (Figure 3D and Appendix A). Concomitantly, a significant difference in mean pixel intensity between hot and cold tumor samples categorized by DigiWest was detected (Figure 3E; Mann–Whitney U test, *p* < 0.05).

When comparing various clinical variables, such as the type of surgery, age and, most notably, the receptor status (ER, PR or Her2), we did not observe any difference between hot and cold breast carcinomas. Importantly, the responder status was the only clinical variable significantly enriched within the hot tumor group (Figure 3F; Table 1; *p* < 0.05, chi-square test).

Tumor infiltration with immune cells is generally associated with patient outcomes and event-free survival (EFS, time from definitive surgery until disease recurrence/metastases or death from any cause). Therefore, we reviewed the difference in clinical outcomes after primary surgery in the present cohort. The group classified as “hot tumors” indeed had a significantly better outcome when evaluating 10-year EFS (Figure 3G; *p* = 0.03, log-rank test). By comparing EFS between hot and cold tumor samples in subgroups categorized through pathological hormone and Her2 receptor statuses, a significant difference was found in ER+ and PR+ samples; yet, more strongly infiltrated samples displayed a tendency toward better EFS (Appendix A). Univariate Cox regression confirmed that the immune cell infiltration status was a prognostic factor of the clinical outcome (*p* = 0.04).

### 3.3. Focused Protein Expression Analysis of Hot and Cold Breast Carcinomas

Next, we performed a detailed analysis to identify differential protein expression in breast carcinomas classified as hot or cold tumors. We allocated the samples to the hot or cold tumor group by assessing immune cell markers as described above and comparing protein expression levels. *n* = 30 analytes displayed a significant difference in expression (Mann–Whitney U test; Benjamini–Hochberg FDR; corrected *p* < 0.05) and a log2 fold change of at least +2/3 or −2/3 (Figure 4A,B,D; see Table 2 for all significantly differentially expressed proteins; Appendix A).

This data analysis revealed an increase in the expression levels of members of the Jak/Stat pathway in the hot tumor subgroup. STAT4, a known mediator of the IL-12 response [27], as well as STAT1, known to be essential for interferon-α (IFN-α) and IFN-γ responses, and its active phospho-variant (Tyr701) [28] were significantly elevated in hot tumors (Figure 4C), indicating the activation of this pathway. Furthermore, Janus tyrosine kinase 2 (Jak2), an important cytokine receptor [29], displayed 3.0-fold elevated expression (log2 fold change of 1.6) in hot tumors. The programmed cell death 1 protein (PD-1), which is known to be an important regulator of immune cell activity [30], was also enriched in this group. Thus, higher immunosurveillance in the hot carcinoma group is characterized by increased expression levels of additional immune cell markers and important members for immune-relevant signaling pathways, supporting the previously established tumor groups.

The expression of CD56, used for the identification of NK cells [31], and IL-2Rα/CD25, characteristic of regulatory T-cells (Tregs) [32], was also increased in this subgroup. Interestingly, the transcription factor Foxp3, which is a common marker for Tregs broadly linked to immunosuppression and tumor protection [33,34], showed significantly increased expression in the cold tumor subgroup (Figure 4C). Looking in detail, our data also showed that CD163, a marker for M2-type tumor-associated macrophages (TAMs), displayed moderately elevated expression levels (0.7-fold log2 increase) in the hot tumor group. While the presence of M2-type macrophages has been linked to tumor progression [35], we interpret this detectable increase as a general indication of higher immune cell activity in this group.

### 3.4. Hot Tumor Samples Show Increased Proliferative Activity and a More Competitive Phenotype

The expression levels of the proto-oncogene tyrosine kinase Src and the calcium channel flower homolog (FLOWER) were found to be increased in hot tumor samples. Src plays a critical role in multiple cellular processes, including proliferation and invasion, and can be a driver of uncontrolled cell growth [36]. The expression of FLOWER, a cellular fitness sensor, has been associated with a competitive growth advantage of cancer cells [37,38]. In addition, the hepatocyte growth factor receptor (c-Met), which is instrumental in increased cell growth and associated with aggressive cancer phenotypes [39,40], was highly expressed in samples with higher immune cell infiltration. Additionally, the transcription factor forkhead box C1 (FoxC1), which is linked to breast cancer invasiveness [41,42], was highly expressed in samples assigned to the hot tumor subgroup. The expression of vascular endothelial cadherin (VE-cadherin), which is important for the adhesion of cancer cells to the endothelium [43], was decreased as compared to the cold tumor group. An increase in vascular permeability by reducing the amount of VE-cadherin via endocytosis has been observed in response to inflammatory activity; this mechanism facilitates immune cell infiltration [44]. Thus, decreased levels of VE-cadherin may be indicative of generally higher immune cell infiltration. Taken together, these results indicate a more competitive phenotype in hot tumor samples.

Additionally, significant promotion of the cell cycle was found in the hot tumor subgroup, as indicated by the higher expression of the important regulatory proteins Cyclin B1 and cyclin-dependent kinase 1 (CDK1), responsible for G2/M-phase progression [45]. This was accompanied by an increase in the phosphorylation of the mitosis marker Histone H3 at Serine 10 [46], as well as a significant decrease in the expression of the cell growth repressor E2F-4 [47].

In conclusion, it is conceivable that the higher expression of proliferative and competitive signaling proteins facilitates the immune recognition of breast cancer cells and therefore leads to higher rates of immune cell infiltration.

### 3.5. Elevated Immune Cell Infiltration Induces Expression of Tumor-Suppressive Markers and Apoptotic Activity

Our data also indicate that hot tumor samples show a significant increase in the expression of several tumor suppressors. The phosphorylation of the tumor suppressor p53 at Serine 20, which enhances p53 activity, leading to cell-cycle arrest or apoptosis [48], was detected, and a 1.2-fold increase in phosphorylation was observed. In addition, increased expression of isocitrate-dehydrogenase 1 (IDH1), another suppressor of tumorigenesis [49], and of phosphorylated and activated MOB1 (pT35), a member of the Hippo pathway [50], was observed in hot tumor samples.

Consequently, this subgroup displayed an increase in apoptotic activity, indicated by the enriched expression of several pro-apoptotic proteins. These include the initiator cysteine aspartic acid protease 9 (caspase 9), serving as an amplifier of the apoptotic response [51], and caspase 6 (one of the major executioner caspases). Both promote apoptosis in its cleaved and active forms [52]. Crucially, we observed the upregulation of the cleaved caspase 9 fragment at 35 kDa as well as the cleaved caspase 6 fragment at 15 kDa (Figure 4C), but not of the full-length proteins in the hot tumor subgroup. Similarly, the expression of the pro-apoptotic factor Bax [53] was significantly enriched in samples with higher immune infiltration. Interestingly, the anti-apoptotic Bcl-2 family member Mcl-1, which antagonizes pro-apoptotic Bcl-2 proteins [54], showed a similar trend. These results suggest that higher immune cell infiltration leads to more apoptotic and tumor-suppressive signaling.

### 3.6. Cold Breast Tumors Show Increased Expression of Immunosuppressive Factors

Our data revealed that the luminal tumor co-marker progesterone receptor (PR), which is linked to an immunosuppressive tumor microenvironment [55,56], displayed a decrease in expression at the highest significance as compared to the cold subgroup (Figure 4B,C). Another contemplated immunomodulation factor is the PPARγ/RXRα pathway, which regulates cell proliferation and inflammation [57]. It has been implied that the expression of peroxisome proliferator-activated receptor γ (PPARγ), a key modulator of this pathway, correlates with suppressed immunosurveillance [58,59,60]. Interestingly, a positive correlation was found between phosphorylated PPARγ (pS112) and PR (Figure 5B, Pearson’s r = 0.45, *p* < 0.0001), indicating the co-expression of both factors by cold carcinomas.

Additionally, our results show that the expression of the phosphorylated variant of PPARγ (pS112), but not the total protein variant, was significantly upregulated in cold tumor samples, whereas CD8 expression displayed the opposite trend (Figure 4C and Figure 5C). Immunohistochemical staining of representative samples verified the predominant expression of PPARγ (pS112) and PR by cancer cells in cold tumor samples (Figure 5A). Concomitantly, the percentage of samples classified as cold tumors was found to be significantly enriched in samples with an elevated (greater than the median) PPARγ-pS112 signal (Figure 5D, Fisher’s exact test, *p* = 0.0046). Hence, our data suggest that PPARγ phosphorylation might be involved in a mechanism governing immune cell repulsion in breast cancer.

## 4. Discussion

Mutational changes in cellular signaling that trigger cell growth are key events in the transformation process that lead to the formation of tumor cells. The targeted analysis of central signaling pathways helps to track the effects of such mutations and can be used to functionally classify tumors by subtype. The DigiWest methodology employed here is capable of achieving this on a much broader scale compared to classical immunohistological approaches, and the knowledge generated by characterizing the activity of central signaling pathways can be utilized to identify novel targets for therapeutic intervention [61].

Here, we used a well-characterized collection of 160 archived breast cancer tissues for targeted protein profiling and aimed to concomitantly detect (i) signaling proteins and their activated variants and (ii) immune cell markers that define the tumor microenvironment. As a result, we were able to screen for aberrations in intra- and extracellular communication and to assign an immune status to each individual tumor tissue. This cohort was chosen since patient-specific follow-up data for all samples were available and could be integrated into the analysis of the correlation between protein expression levels, immune status and tumor reoccurrence. Yet, a detailed analysis of the prepared tissue sections revealed that the tumor content in a significant fraction of the archived tissues was lower than 50% (75/159, corresponding to 47%). Since only samples with more than 50% tumor content were analyzed, the number of patient samples was reduced to 84. Therefore, some care has to be taken when interpreting the obtained results.

Immune cell infiltration of breast cancer tissue and stroma is linked to a better prognosis [6]. On the contrary, non-immune-infiltrated tumors show no or a low response to current immune therapy [17]. To survey for the presence of infiltrating immune cells in cancer tissue, we measured central immune cell markers (CD8a, CD11c, CD16 and CD68) simultaneously and categorized the present cohort into highly infiltrated (“hot”) tumor and lowly infiltrated (“cold”) tumor samples. The assessment of patient outcome data revealed a significant difference in event-free survival in favor of the highly immune-cell-infiltrated group. Significantly, higher amounts of PD-1 and additional immune cell markers were detected in these samples, indicating higher immunosurveillance. Conversely, the specific Treg cell marker FoxP3, necessary for immune-suppressive activity and immunological tolerance [62,63,64], was found to be enriched in cold breast tumors. This indicates that in this cohort, higher expression of FoxP3+ cells leads to the retention of immune cell infiltration within the tumor tissue and, consequently, poorer patient outcomes.

At the same time, we show that highly immune-cell-infiltrated tumor samples display elevated immunological signaling activity, as was indicated by the upregulation of regulatory proteins Jak2, STAT4 and STAT1, including its activating phosphorylation at Tyrosine 701. These crucial members of the Jak/Stat pathway are important for the cytokine response and constitute key regulators of the immune system [65,66].

In a recent study, high apoptotic activity in breast cancer tissue was shown to be associated with the high infiltration of immune cells. Based on mRNA expression data, the authors hypothesized that increased apoptosis is associated with immune cell killing [67]. Here, we report that in hot breast cancer tissue, central apoptotic marker proteins such as the cleaved version of the initiator caspase 9, the cleaved effector caspase 6 and Bax were upregulated. Increased levels of p53 phosphorylated at Ser20 and Histone H3 phosphorylated at Ser10 in these tumors indicate the likely involvement of DNA damage [48].

Phosphorylated PPARγ (pS112) was present in higher amounts in samples with lower immune cell infiltration. Besides its common function in adipogenesis and lipid metabolism [68], PPARv has been linked to worse outcomes in breast cancer patients [69,70], as well as the evasion of immunosurveillance and the impairment of CD8 T-cell infiltration in muscle-invasive bladder cancer [71]. Our data suggest that higher PPARγ phosphorylation impairs immune cell infiltration in breast cancer, ultimately worsening patient outcomes. Therefore, we hypothesize that PPARγ phosphorylation may be involved in an immunosurveillance evasion mechanism employed by breast cancer cells. Hence, PPARγ and its phosphorylated form (pS112) may serve as potential markers for patient stratification or as a target for therapeutic intervention. However, additional research is required to elucidate this question further.

The application of DigiWest technology enabled us to review the receptor status, immune cell infiltration and protein expression of approx. 150 proteins and protein variants in parallel from minimal amounts of fresh-frozen breast cancer biopsies, demonstrating the unique potential held by this approach.

## 5. Conclusions

In the present study, we investigated cellular signaling and immune cell infiltration in a cohort of 84 breast patients not under a neoadjuvant treatment scheme. Tumor resections were analyzed for the expression of cellular signaling molecules and immune cell markers by semi-quantitatively measuring more than 150 proteins employing DigiWest technology. Based on the obtained expression profiles, the primary tumor tissues were categorized into immunological cold or hot tissues (presence of CD8α, CD11c, CD16 and CD68); the impact of immune cell infiltration on event-free survival was established. This differential expression analysis showed that hot tumor samples displayed higher levels of immunological signaling as well as high apoptotic activity. Elevated immunological signaling was indicated by the activation of Jak/STAT signaling, which was seen at different points of the signaling cascade. Notably, the phosphorylation of PPARγ at Serine 112 was found to be predominant in immunologically cold tumors, indicating its involvement in an immune surveillance evasion mechanism in breast cancer.

These distinct differences in the tumor microenvironment and in intracellular signaling in the tumor cells point to the modulation of intracellular signaling only in tumors that do not show invasion by immune cells. Further knowledge on the underlying mechanism will be useful for identifying patients that show a high probability of relapse, ultimately improving diagnosis, treatment and patient outcomes.

## Figures and Tables

**Figure 1 cancers-14-04542-f001:**
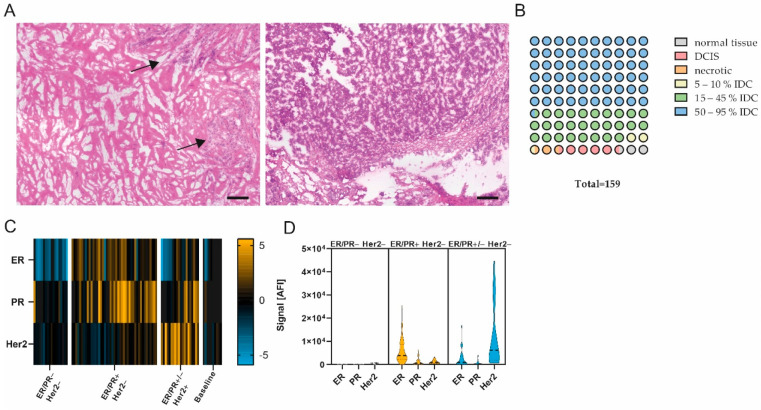
Pathological examination and quality assessment of selected samples. (**A**) Representative images of H&E-stained breast carcinomas. (Left) App. 20% tumor content. (Right) App. 90% tumor content. Scale bar, 200 µm. Black arrows, tumor area. (**B**) Pathological classification of entire sample set (1 circle = 1%). *n* = 159. (**C**) Heatmap showing ER, PR and Her2 protein expression in ER/PR− Her2− (*n* = 18), ER/PR+ Her2− (*n* = 45), ER/PR+/− Her2+ (*n* = 20) and baseline (*n* = 10) subgroup; data are median-centered and log2-transformed. Yellow indicates higher expression; blue indicates lower protein expression. (**D**) Violin plots of ER, PR and Her2 expression in ER/PR− Her2−, ER/PR+ Her2− and ER/PR+/− Her2+ subgroups.

**Figure 2 cancers-14-04542-f002:**
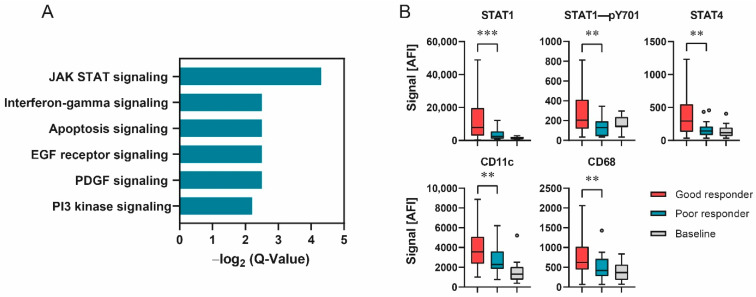
Responder status depended on differences in signal transduction. (**A**) –Log2 Q-Values of PANTHER pathway analysis as bar graphs. (**B**) Protein expression of selected analytes displaying significant differences in expression between responder groups. Data shown as box–whisker plots for good-responder (*n* = 58), poor-responder (*n* = 21) and baseline (*n* = 10) subgroups. *** *p* < 0.001, ** *p* < 0.01. Mann–Whitney U test.

**Figure 3 cancers-14-04542-f003:**
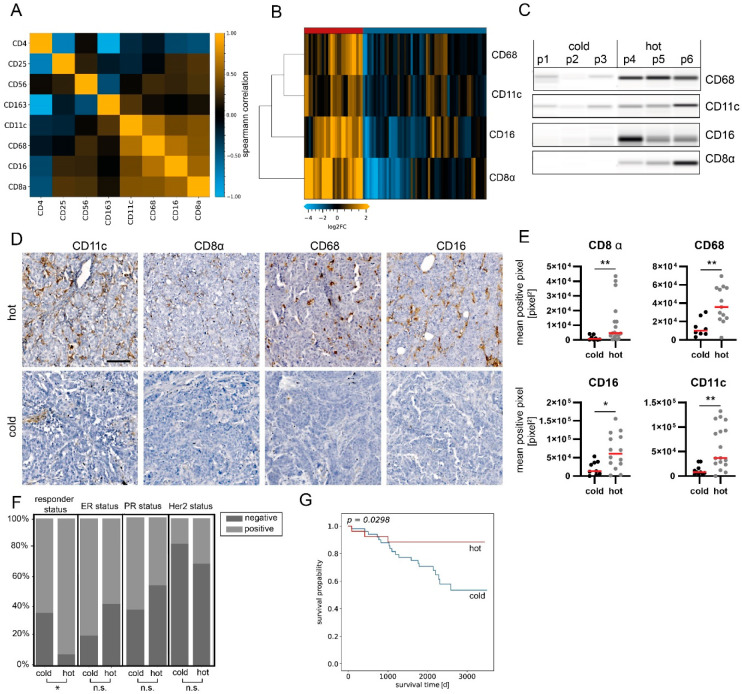
Sample stratification based on immune marker assessment. (**A**) Correlation plot (Spearman’s correlation) of immune cell marker expression in the analysis sample set. The highest correlation was found for CD8α, CD11c, CD68 and CD16 (r > 0.55). *n* = 84. (**B**) Heatmap showing protein expression levels of CD8α, CD11c, CD68 and CD16. Hierarchical clustering of analytes and samples with Euclidean distance and complete linkage. Data are normalized to total protein, centered on median of all samples and log2-transformed. (**C**) Representative Western blot mimics of CD8α, CD11c, CD68 and CD16 (grayscale maps generated from DigiWest data). For graphical representation, background-subtracted raw data from representative hot and cold samples were used. *n* = 3. The uncropped blots are shown in Appendix A. (**D**) Representative images of CD11c, CD8, CD68 and CD16 immunohistochemical staining in hot and cold samples. Scale bar, 50 µm. (**E**) Mean positive pixel values of 5 representative 10× sections per available FFPE sample for hot and cold carcinomas classified by DigiWest. CD8α: cold *n* = 8, hot *n* = 17; CD68: cold *n* = 7, hot *n* = 13; CD16: cold = 9, hot *n* = 14; CD11c: cold *n* = 10, hot *n* = 17. Mann–Whitney U test, ** *p* < 0.01; * *p* < 0.05. (**F**) Distributions of responder, ER, PR and Her2 statuses as percentages, stratified by infiltration status. Chi-square-test, * *p* < 0.05; ns indicates no significant difference. (**G**) Kaplan–Meier analyses of event-free survival in patients stratified by infiltration status. *p* = 0.0298, log-rank test.

**Figure 4 cancers-14-04542-f004:**
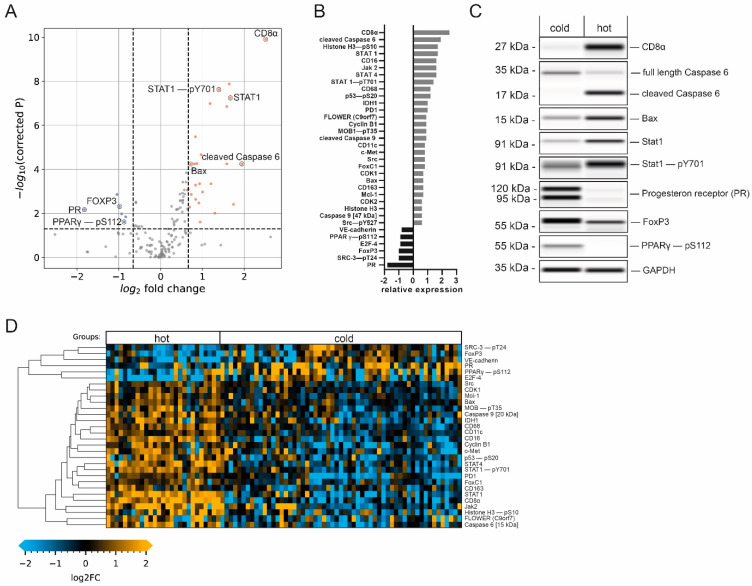
Immune-cell-infiltration-dependent changes in protein expression. (**A**) Volcano plot (−log10-corrected *p*-value versus log2 fold change) depicting differences in protein expression between hot (*n* = 27) and cold (*n* = 57) samples. Data from 150 proteins and protein variants were analyzed. Wilcoxon rank-sum test with Benjamini–Hochberg multiple testing correction; the horizontal dashed line indicates a *p*-value of 0.05; the vertical dashed line indicates a log2 fold change of at least 2/3. Blue and red dots indicate analytes with at least a 2/3-fold difference in median expression and a *p*-value below 0.05. (**B**) Bar graphs of log2-transformed ratios calculated from the mean protein expression in samples with high and low infiltration. Analytes are sorted from the largest positive change to the largest negative change. (**C**) Western blot mimics of selected analytes from two representative patients (grayscale maps generated from DigiWest data). For graphical representation, background-subtracted raw data were used. The uncropped blots are shown in Appendix A. (**D**) Heatmap of analytes with significantly different expression between hot and cold samples displaying a fold change greater than 2/3. Hierarchical clustering of analytes using Euclidean distance and complete linkage.

**Figure 5 cancers-14-04542-f005:**
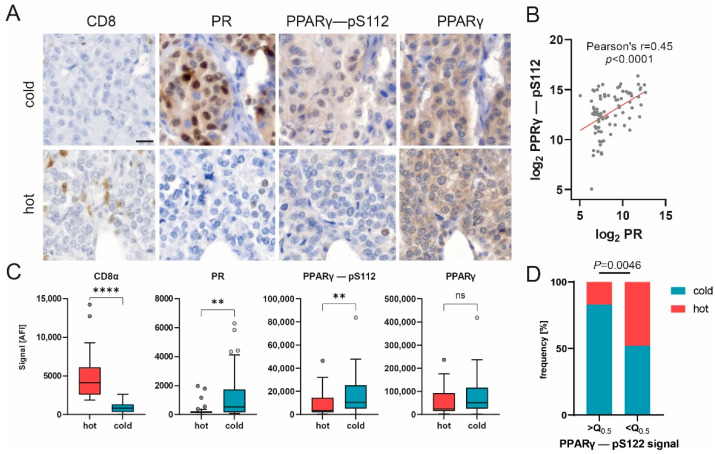
PPARγ phosphorylation correlates with PR expression and may impair immune infiltration. (**A**) Representative images of CD8, PR, PPARγ-pS112 and PPARγ immunohistochemical staining in hot and cold carcinomas. Scale bar, 20 µm. (**B**) Scatter plot of log2-transformed DigiWest signal for PPARγ-pS112 and PR. Pearson correlation, r = 0.45, *p* < 0.0001. (**C**) Box plots showing protein expression for CD8α, PR, PPARγ-pS112 and PPARγ in cold (*n* = 57), hot (*n* = 27) and baseline (*n* = 10) samples. Mann–Whitney U test, ** *p* < 0.01, **** *p* < 0.0001; ns indicates no significant difference. (**D**) Distribution of hot and cold statuses in carcinoma samples with PPARγ–pS112 expression higher or lower than median expression. *n* = 42. Q0.5 = 7004 AFI. Fisher’s exact test, *p* = 0.0046.

**Table 1 cancers-14-04542-t001:** Patient and tumor characteristics of all samples included in the analysis sample set, stratified by responder and immune infiltration status, as indicated by DigiWest. *p*: Chi-Square or Wilcoxon rank-sum test between the two groups.

Characteristic	Overall (*n* = 84)		Poor Responder (*n* = 21)		Good Responder (*n* = 58)			Cold (*n* = 57)		Hot (*n* = 27)		
	No. of Patients	%	No. of Patients	%	No. of Patients	%	*p*	No. of Patients	%	No. of Patients	%	*p*
**Follow-up, years**							0.49					0.7
Median (range)	6.1 (0.2–9.9)		6.3 (0.2–9.9)		6.2 (0.3–9.6)			6.3 (0.2–9.9)		6.2 (0.9–9.4)		
**Age at surgery, years**							0.13					0.4
Median (range)	61 (30–85)		66 (41–85)		58 (84–30)			60 (31–85)		62 (30–84)		
**Tumor size (cm)**							0.15					0.3
<2	23	27.4	4	19.0	17	29.3		14	24.6	8	29.6	
2–5	55	65.5	13	61.9	39	67.2		37	64.9	19	70.4	
>5	6	7.1	4	19.0	2	3.4		6	10.5	0	0	
**Nodal status**							0.4					0.3
Negative	47	56.0	10	47.6	35	60.34		29	50.9	18	66.7	
Positive	36	42.9	11	52.4	22	37.93		27	47.4	9	33.3	
Unknown	1	1.2	0	0.0	1	1.72		1	1.8	0	0	
**Hormone receptor status**												
ER-positive	24	28.6	15	71.4	42	72.4	0.84	44	77.2	16	59.3	0.2
ER-negative	60	71.4	6	28.6	16	27.6		13	22.8	11	40.7	
PR-positive	38	45.2	12	57.1	33	56.9	0.81	34	59.6	12	44.4	0.3
PR-negative	46	54.8	9	42.9	25	43.1		23	40.4	15	55.6	
**HER2 status**							0.83					0.5
Positive	20	23.8	4	19.0	14	24.1		11	19.3	8	29.6	
Negative	63	75.0	17	81.0	43	74.1		45	78.9	18	66.7	
Unknown	1	1.2	0	0.0	1	1.7		1	1.8	1	3.7	
**Type of surgery**							0.27					0.4
BCS	47	56.0	12	57.1	34	58.6		34	59.6	13	48.1	
SSM	1	1.2	1	4.8	0	0.0		1	1.8	0	0	
Ablatio	16	19.0	3	14.3	10	17.2		5	8.8	5	18.5	
Mastectomy	10	11.9	5	23.8	5	8.6		2	3.5	0	0	
Quadrantectomy	2	2.4	0	0.0	1	1.7		10	17.5	5	18.5	
Segmental resection	1	1.2	0	0.0	1	1.7		0	0	1	3.7	
Mastopexy	5	6.0	0	0.0	5	8.6		3	5.3	2	7.4	
NSM	1	1.2	0	0.0	1	1.7		1	1.8	0	0	
Unknown	1	1.2	0	0.0	1	1.7		1	1.8	0	0	
**Responder status**												0.02
Poor responder	21	25.0	-	-	-	-		19	33.3	2	7	
Good responder	58	69.0	-	-	-	-		34	59.6	24	89	
Unknown	5	6.0						4	7.0	1	4	

**Table 2 cancers-14-04542-t002:** Protein analytes displaying significantly different expression between hot and cold breast tumor samples (Mann–Whitney U test; Benjamini–Hochberg FDR; corrected *p* < 0.05).

Analyte	Uncorrected *p* Value	Corrected *p* Value	log2 Fold
PR	0.002	0.007	−1.8
SRC-3—pT24	<0.001	0.001	−1
FoxP3	0.001	0.005	−1
E2F-4	0.003	0.011	−0.9
PPAR gamma—pS112	0.009	0.025	−0.9
VE-cadherin	0.005	0.015	−0.8
Cytokeratin 8/18	0.016	0.040	−0.6
Glycogen Synthase—pS641	0.014	0.036	−0.6
PTEN	0.001	0.003	−0.6
PDK1	0.001	0.003	−0.6
PTEN—pS380	0.017	0.041	−0.6
mTOR	0.016	0.040	−0.5
Dvl2	0.016	0.040	−0.4
MAD2L1	0.001	0.005	0.3
E2F-1	0.006	0.018	0.3
p70 S6 kinase—pT389	0.008	0.024	0.3
Erk1/2	0.011	0.030	0.3
IKK alpha	0.013	0.035	0.3
p38 MAPK—pT180/Y182	0.008	0.023	0.4
PD-L1	0.002	0.007	0.4
A-Raf	0.002	0.007	0.4
TSG101	0.007	0.020	0.4
NF-kB p65—pS468	0.002	0.006	0.4
PD-L1	<0.001	0.001	0.5
NF-kB p105/p50 [50 kDa]	0.001	0.003	0.5
CD25	0.001	0.003	0.5
c-Raf—pS259	0.002	0.006	0.5
PI3-kinase p85	0.003	0.009	0.5
CD56	<0.001	0.001	0.5
p38 MAPK	0.001	0.003	0.5
GSK3 beta	0.001	0.003	0.5
RUNX2	<0.001	0.000	0.5
p53—pS37	0.001	0.003	0.5
STAT 5 alpha	<0.001	0.002	0.5
MEK2	<0.001	0.001	0.6
Caspase 3	<0.001	0.001	0.6
Src—pY527	0.008	0.024	0.6
Caspase 9 [47 kDa]	<0.001	<0.001	0.6
Histone H3	<0.001	<0.001	0.6
CDK2	0.002	0.007	0.6
Mcl-1	<0.001	0.001	0.7
CD163	0.001	0.005	0.7
Bax	<0.001	<0.001	0.7
CDK1	<0.001	0.001	0.7
FoxC1	<0.001	<0.001	0.8
Src	<0.001	0.001	0.8
c-Met	<0.001	0.002	0.8
CD11c	<0.001	<0.001	0.8
Caspase 9 [35 kDa]	<0.001	<0.001	0.9
MOB1—pT35	0.001	0.003	0.9
Cyclin B1	<0.001	<0.001	0.9
FLOWER (C9orf7)	0.009	0.025	0.9
PD1	<0.001	<0.001	1
IDH1	<0.001	0.001	1
p53—pS20	<0.001	<0.001	1.2
CD68	<0.001	<0.001	1.2
STAT 1—pT701	<0.001	<0.001	1.4
STAT 4	<0.001	<0.001	1.6
Jak 2	<0.001	<0.001	1.6
CD16	<0.001	<0.001	1.6
STAT 1	<0.001	<0.001	1.7
Histone H3—pS10	0.001	0.004	1.7
Caspase 6 [15 kDa]	<0.001	<0.001	1.9
CD8a	<0.001	<0.001	2.5

## Data Availability

The data that support the findings of this study are available from the corresponding authors upon reasonable request.

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
