# Peer review of "Protein Profiling of Breast Carcinomas Reveals Expression of Immune-Suppressive Factors and Signatures Relevant for Patient Outcome"

_cancers, 2022, doi:10.3390/cancers14184542_

Round 1

Reviewer 1 Report

It is known that in cancer, the complex interaction of tumor cells and the tumor microenvironment leads to the modulation of signaling processes. By evaluating the expression of a variety of proteins and protein variants in cancer tissue, a wealth of information about signaling pathway activation and the state of the immunological landscape can be obtained, which can provide valuable information for treatment response. The authors of this article used archived breast cancer tissue from 84 patients and analyzed it using high-throughput Western blotting, and examined the expression of 150 proteins covering the central cancer pathways and immune cell markers. By evaluating the expression of CD8α, CD11c, CD16 and CD68, immune cell infiltration was determined and a strong correlation of event-free survival of patients with immune cell infiltration was found. The presence of tumor-infiltrating lymphocytes was associated with pronounced activation of the Jak/Stat signaling pathway and apoptotic processes. Increased PPARγ (pS112) phosphorylation in non-immune infiltrated tumors suggests a novel immune evasion mechanism in breast cancer characterized by increased PPARγ phosphorylation. Multiplex evaluation of immune cell markers and profiling of tumor tissue proteins provide functional signaling data to facilitate stratification of breast cancer patients.

I liked the article, the clear and logical design of the study, adequately selected illustrative material that does not overload the article with data. Among the minor remarks is a very brief introduction, which can be expanded, since the tumor microenvironment is studied quite extensively.

Reviewer 2 Report

This manuscripts uses a well characterised breast cancer archive to assess the individual tumor's immune responses. Tumors are sectioned and sections stained for classical histology, immune stained and used for DigiWest protein analysis (levels and activities). The western data are related to patient outcome, responsiveness to treatment and other tumor characteristics (ER, PR, Her2, IC infiltration). The paper is solid, well presented and of interest, although most of the IC data are not very surprising based on the literature. A novel twist is PPARg involvement. Of course, more IC subtypes could be studied for a more complete assessment.

I have only a few minor comments that can be addressed. Page 11, CD163. A M2 marker is increased in the hot group (which has a better survival). This is an apparent contradiction since M2 macrophages are considered pro-tumoral and should be better discussed.

The discussion on VE-cadherin does not impress since VE-cadherin is primarily an EC marker that regulates vascular permeability and that avenue should be opened in the reference to these data. Thus, the changes in VE-cadherin could reflect vascular density and/or leakage and these parameters may influence the immune response. A recent review addresses this possibility (https://doi.org/10.3390/ ijms23042313).

Typos: supress, VE-cadherine (Suppl data)
